# Local's attitude towards African elephant conservation in and around Chebra Churchura National Park, Ethiopia

**Adane Tsegaye** *, **Afework Bekele, Anagaw Atikem**

Departments of Zoological Sciences, Addis Ababa University, Addis Ababa, Ethiopia

* adanetsegaye263@yahoo.com

**Data Availability Statement:** All relevant data are within the paper and its Supporting Information files.

## Abstract

Economic growth and development in developing countries often involves land-use changes which fragment natural areas, bring humans and wildlife into closer proximity and escalating human- wildlife conflicts. Human-wildlife conflicts impose huge costs on local people and their livelihoods. Balancing developmental activities with the conservation of mega fauna such as the African and Asian elephants (*Loxodonta Africana*, *Elephas maximus*; respectively) remains problematic. Understanding the reasoning upon which perceived risks and level of human- elephant conflict laid is critical to address societal or cultural beliefs in order to develop effective mitigation strategies. The perceived risks and level of conflict have to be properly addressed for effective planning and implementation of appropriate mitigation strategies. We studied human- elephant interactions in Chebra Churchura National Park Ethiopia (CCNP) from September 8 to October 28, 2022 and collected baseline data on human perceptions of conflicts in an area where elephant populations are increasing. To complete our study, we surveyed 800 household from 20 villages adjacent to the CCNP. The purpose of this investigation was to identify the relevance of the existing human-elephant conflict (HEC) with the attitude of local communities towards elephant conservation, the park management and perceived effective mitigation techniques. The local communities trust in the implementation of different traditional mitigation techniques is generally weak. The households interviewed were less positive towards the effectiveness of most of the traditional techniques in chasing elephants away from their farm lands. They believed that elephants had already adapted and do not respond to most of these techniques. Against the above perception in exception of their usual absence and late arrival, perception of local communities about shooting warning gun by park scouts is among the most accepted effective methods in chasing elephants from their farm lands. The majority of respondents believe that separation of elephants and humans by constricting barriers is the best solution to the problem. The idea of constructing barriers such as electric fence; ditch or concrete wall and blocking corridors between the Park boundary and the villages have become the most popular idea of local communities followed by relocating people to other safer places, as the best protection method against the elephant attack irrespective of the associated initial and maintenance costs.

**Funding:** Funded studies Funded by Rufford Small Grant Grant Number: ID:39147-2 Grant Awarded to the corresponding Author ADANE TSEGAYE TEGEGNE The funders had no role in study design, data collection and analysis, decision to publish, or preparation of the manuscript.

**Competing interests:** No authors have competing interests

# Introduction

According to (24). developmental activities in developing countries often involve alteration of natural wildlife habitats into agricultural farm lands, human settlements and industries areas. Altering natural lands for such activities often leads to fragmentation and loss of natural habitats. Such activities finally end up in resulting humans and wildlife living in closer proximity and escalating human–wildlife conflict (HWC) [1–4]. Loss of natural habitats to expand developmental activities such as agricultural farm lands, industrializations, settlements and urbanizations to meet the ever growing demand of human populations has become one of the top conservation challenges especially for the mega fauna such as elephants. HWC imposes huge cost on local people and their livelihoods [5–11]. More over the expansion of these activities resulted in widespread habitat loss, fragmentations and loss of landscape connectives across Asia and Africa followed by huge decline in the elephant populations from most of their previous natural habitats and ranges [3, 12–17]. Currently most of the elephant populations in Asia and Africa are forced to live in close proximity with human due to a significant loss of their habitats, resulting competitions for space and resources with people and Sever conflicts with human beings including crop raid, injury and death of domestic animal and human beings [12, 18, 19]. Negative interaction between human and elephants have become known as HEC. HEC has been identified as one of the five issues having equal priority that needs attention regarding the African elephant conservation African Elephant Specialist Group [3, 20–22]. In Africa only 20 percent of the species range has any form of protection but conflicts occur at almost any interface [20, 22]. More over the issue is becoming increasingly politicized locally, even if actual incidents are sporadic or limited impact [20, 22]. Thus, understanding the elephants actual impact versus perceived level of conflict, local livelihoods and household production are valuable and timely to identify the principal driving causes of the conflicts and investigate and develop effective conflict prevention and mitigation measures [23–26].

The African elephant population in CCNP, Southwestern Ethiopia is increasing and there are concerns for escalating conflict with the local community. Conflicts which are not properly addressed and managed are the major causes of poor management and conservation of many wildlife species and protected areas [9, 27, 28]. To design and implement efficient and sustainable conflict mitigation strategies and plans, ensuring the willingness and full participation of local communities is mandatory for its success and sustainability [29–31]. Thus, needs and concerns of local communities must be addressed and incorporated in the process. The level and consequences of HEC varies between localities depending on means of lively hoods, crop type grown, environmental conditions and habitat characteristics [5, 32, 33]. Based on the intensity and frequency of the above factors, perceptions of local communities towards HEC, conservation of elephants and park management vary between localities [33]. To address the HEC issues properly, it requires full understanding of the complex nature of human-elephant conflict and its driving factors that are basis for planning and implementation of effective mitigation techniques and management plans [31].

Understanding local community's perceptions and attitudes towards HEC and conservation of African elephants is vital to investigate, plan and implement mitigation techniques with full participation of local communities. Studies confirmed that benefiting local communities from income generated from protected areas, ecological services provided by wildlife and the recreational values of wildlife protected areas are the main reasons for tolerance to human- wildlife conflict and long-term co-existence between wild animals and local communities living in close proximity with wildlife protected areas [34]. However, against this fact frequent conflict with particular animal species may aggravate negative perception about the animal within local communities, making the long term co-existence difficult with the species concerned.

In addition to this important point, socio economic variables such as income level, gender, age categories and occupations are among the factors highly affecting people perceptions towards wildlife conservation and the species concerned [34]. Managing conflicts is especially challenging for large, potentially dangerous and damaging species such as elephants [28]. This study aimed to determine local community's perception about human-elephant conflict and conservation of African elephants, the relationship between the conflicts and local livelihoods, household productions and local people's perceptions about conflict prevention and mitigation measures. Our study provides an important baseline data for developing appropriate conservation management plan to resolve the HEC. A good knowledge and understanding of community perceptions of HWC, their attitude towards the particular species and its conservation is vital to ensure long-term co-existence between the species and local communities living in close proximity with the wildlife habitats. Interviews with local people may help to uncover the main factors associated with the conflicts, their spatial uniqueness and commonalities [32, 35–37]. To investigate the relationships between livelihoods of local communities, their perceptions of conflicts and attitudes towards elephant conservation, we conducted interviews to 800 household in 20 park adjacent villages. Our objectives were 1. To investigate if human-elephant conflicts exist, their main causes and types in different villages. 2. Assess general attitudes of local communities towards elephant conservation and the park management. 3. To examine the attitudinal difference towards elephant conservation and the park management among age and sex groups, occupations and educational levels. 4. To identify effective HEC mitigation techniques those are trusted by local communities and can be implemented locally. The overall purpose of this paper is to improve understanding of the variables that influence attitudes toward African elephant conservation, with a goal of mitigating conflicts and promoting human-elephant coexistence. We also investigated several hypotheses regarding the relationship between socio-economic variables and perceptions of elephants in the area and how socio- economic variables such as; gender, age, educational level, income and occupation affect the local communities perception of elephants. We predicted that socio-economic variables, as well as past experiences with elephants, would largely affect local community feelings toward elephant conservation. We explored perceptions of HEC for 800 households distributed in 20 villages around CCNP, examining trends in crop raid, damage to human and domestic animals, people's attitude towards elephants and their conservation. We also investigated local people's perceptions about the preferred methods for reducing HEC. We hypothesized that the existence, frequency and intensity of HWC varies among the village. We hypothesized the local communities located adjacent to the park boundary identify HEC as a major challenge that affect their lives and livelihoods. We hypothesized that farmers experience more HEC due to crop raiding by elephants than non-farmer community members. We hypothesized that people who experienced frequent conflict perceive elephants to be problematic than people in villages that did not experience conflict. Finally, we expected that people who experienced higher levels of HEC express lower levels of support for elephant conservation and the park management.

## The study area location

Chebera Churchura National Park Ethiopia, (CCNP) is located in southwestern part of Ethiopia, in the newly established Southwest Ethiopia Administrative Region (SWER). The Park is located between Dawro and Konta Zones at about 233 km and 475 km southeast of Bonga (the capital city of SWER and south west of Addis Ababa (the capital city of Ethiopia) respectively. It covers an area of 1410 km$^2$ and lies between the coordinates 36° 27'00"- 36° 57'14"E and 6° 56'05"-7° 08'02"N. Bordered by Konta Zone to the north, Omo river to the south, Dawro Zone to the east and southeast and Agare High mountains and Ouma river to the west [4, 38].

The area has two main seasons the wet and dry season with uniform and long rainfall season (between March and September and with a peak in July) representing the wet season. The dry season of the area is from November to February, with mean maximum temperature varying between 27 and 29°C.

The hottest months are January and February while, the coldest months are July and August with the mean maximum and minimum temperatures of 28°C and 12°C, respectively [39, 40]. The area is characterized by a unique and highly heterogeneous hilly terrain with elevations ranging between 550 to 2450 masl. Large portion of the area is highly undulated interspersed with different valley floors, drained bottomland with different hills and lies at the center of the Omo-Gibe river basin. It contains both the highland and lowland of the basin.

The highlands are characterized by steep slopes. The lowlands, by contrast, are characterized by low altitude and relatively gentle slope [4, 41]. The principal ethnic groups found around CCNP are Dawro and Konta Nationalities. Other minority groups include Tsara, Menja, Mena and Bacha. Dawro ethnic group inhabits the eastern highland and few areas of the southwestern lowland areas. These people do not make extensive use of the lowlands except along the periphery. Konta ethnic group occupies the north and northwestern highland areas. Churchura farmers inhabit the southern lowland. Mixed agricultural practices are the sole livelihood of the majority of the inhabitants around the Park area. The people practice traditional agricultural system that combines perennial and annual cultivation with livestock rearing. Permanent crops harvested in the area include cereals, fruits, enset and vegetables. Enset, sorghum and maize are the major staple crops, and mainly used for household subsistence. Coffee and honey are the major income earning products of the area [42].

A wide range of fruits and vegetables are also cultivated both for subsistence and for sale. Teff is cultivated mainly for cash [41]. The minority groups of people also lead their livelihood by collecting and selling wild honey, spices, and wild coffee and edible roots from the forests of some of the wild plants [42].

CCNP is known to possess high diversity of flora and fauna. So far, 40 large and medium sized mammals including four of the five big game animals, 18 species of small mammals of which one is endemic to Ethiopia and 138 species of bird of which 6 are endemic to Ethiopia are recorded in this National Park. Two mammalian species, Weyns' duiker (*Cephalophu sweynsi*) and Harvey's duiker (*C. harveyi*) were also recently recorded in this National Park which was not officially recorded in Ethiopia before. The Park is also the only home in for an endemic species of fish "*Gara chebra*" which is named after the National park. 106 woody plant species were, of which 6 (*Millitiaferugeni*, *Vepris daneli*, *Solanecio gigas*, *Cussonia ostini*, *Erythrina brucei*, *Rhusglutinosa*) are endemic to Ethiopia [8].

The vegetation cover of the area is categorized in to four main types; wooded grassland, woodland, montane forest and riparian forest. Wooded grassland accounts for 55.6% of the study area. It covers most of the undulating landscapes above the floor of the valleys and gorges. Although the grass species show local variation, the dominant grass species is the elephant grass *Pennisetum* sp [43]. The tree species are deciduous and include *Combretum* sp. in association with *Terminalia albiza*. Woodland habitat covers about 13.2% of the total area while the riparian forest habitat covers only 3% of the total area of the Park. The montane forest habitat covers about 27.2% of the total area of the Park.

## Methods

We conducted our study in villages adjacent to the CCNP that had the greatest potential for human-elephant conflicts. We used in-person self-administered questionnaires and focus group discussions by modifying [42, 44]. Data were collected from a total of 20 Park adjacent

villages during the survey. The study was aimed to assess the level of HEC, the attitude of local communities towards conservation of elephants and the park and possible conflict prevention and mitigation measures that can be implemented by local communities. Procedures were followed according to relevant laws and guidelines of the country. Permission was given by all the concerned institutions. Before the actual data collection, the study methods were well examined and approved by all concerned governmental institutions including the Ethiopian Wildlife Conservation Authority, the Regional Tourism Bureau, CCNP Office and local administrator. Participatory discussions based on full willingness of the respondents were made. Prior to data collection, we completed a reconnaissance survey during July 2022, we collected general information about the park, local community's livelihoods, elephant spatial and temporal habitat-use patterns and human-elephant interactions.

We visited each house hold in the 20 park adjacent villages to interview adults who were available and willing to participate in the study at the time. We conducted interviews with both men and women between the age of 18 and 79 years, from September 8 to October 28, 2022. Each interview lasted 10 to 30 minutes and responses to the questions (Appendix A) were recorded on a survey form. Our questionnaires were mainly focused on six main areas; (1) crop types commonly grown and palatable for the elephants. (2) types of conflict (3) elephant population trends and perceptions of local communities about elephant conservation and conflicts, (4) the perceived effective conflict prevention and mitigation measures, (5) people's attitudes towards future conservation of the Park and elephants, (6) The best mitigation techniques that are believed to be effective by local communities and to be implemented in the future to solve human-elephant conflict problem. We administered the structured questioners to the member of household at a random manner excluding household members' age less than 18 based on first come first served biases [42, 44].

We identified 15% of the 20 park adjacent villagers for follow-up interviews and focal group discussion. Our focus groups discussions were also conducted in the villages to discuss the experience of people in human elephant conflicts and the effectiveness of different possible mitigation and prevention measures implemented by the local communities.

## Questionnaire design

After recording socio demographic data (gender, age, and occupation), we asked participants, based on our objectives. To investigate the spatial patterns of the conflict we asked participants "Did you ever experience any form of conflict with elephants?" Ex, crop raid or loss and injuries of human and domestic animal Responses were categorized into the following: (a) yes, (b) No. If the response to question number 1 is no and the existence of HEC was not confirmed, the participants were not asked the next question. If the response to question no 1 is yes then the respondent will be asked the following serious of questions. We tested responses for significance between the existence and absence of conflicts in different villages using (chi-square test, SPSS software version 20). Respondents who says yes for question number 1 then will be asked "What is/are your source of income?". Responses were categorized into the following: (a) Crop farming (crop cultivation) b. livestock rearing c. mixed farming both (Crop farming and livestock rearing). d. Other occupation such as laborer e. Students or jobless f. Others (mention). We tested responses for significance between farmers' and non-farmers' responses using (chi-square taste, SPSS software version 20).

To identify the major reasons and impacts of human-elephant conflict that they experienced we asked, "What are the main causes of conflicts you have with elephants?" response categories were the following: (a) crop raiding, (b) human injury, (c) domestic animals injury, (d) loss of human lives, (e) others (mention). To identify if they believe/experienced that some

crop types are preferred and others are avoided by elephants so that they can farm crops that are less preferred by elephants, we asked the participants to list out the crop types commonly cultivated in their village and palatability/preference to elephants. Response recorded were as follow: Lists of crop types commonly cultivated in their villages first and then they were asked to identify each crop type they listed as palatable or preferred by elephants or non-palatable as; a. palatable b. non-palatable. To identify specific elephant groups causing most of the conflicts participants were asked to answer a question as follows; a. mixed herds b. bull groups c. lonely bull. To investigate the temporal patterns of HEC participants were asked about the specific time of attack by elephants. Responses were categorized into the following: a. Morning b. Afternoon c. Evening d. Night.

To investigate if human death and injuries caused by elephants were the reason for their current perception of elephants. The participants were asked open questions to explain if any one injured or killed by elephants from their family in the last two years. Response were categorized as; a. Injury b. Death. Participates were then asked to explain about a. Specific date of the incidence b. Specific location of the incidence. To evaluate if elephants have impact on another income source, domestic animals, the participants were asked to explain if elephants had ever attacked their domestic animals in the last two years. If their response is yes then they were asked to indicate type of domestic animal as follow; a. Cattle b. sheep c. goat. D. other (specify). We also asked the estimated monetary value of the animal lost based on local market price. Response were categorized as follow'.1. What incidence/attack have you ever faced concerning your domestic animal in the last two years a. Injury b. Death? They were also asked to indicate. A. The specific date of the incidence. b. The specific location c. The estimated monetary value of the animal killed at local markets. To assess the local communities perceptions of elephants and their attitudes towards elephant conservation and the park, we asked community members if it was important to conserve elephants, the park and to have elephants in the future around their village. If their response was yes, we asked them to indicate either yes or no on why they are important: (a) because they are part of nature, (b) for religious reasons, (c) for tourism, or (d) for other reasons. We also asked two multiple-choice questions to determine whom they believed should be responsible for HEC management, and if the participant, personally, would be willing to contribute to HEC mitigation initiatives. We also asked two open-ended questions to identify what actions participants were currently taking to prevent elephant damage. Respondents were asked open ended question to list different techniques, sequentially based on their level of effectiveness and availability in their localities they implement to minimize the problems caused by elephants on cropland at night. Finally we asked them to indicate the best mitigation techniques to be implemented in the future to solve HEC problem.

## Data analysis

Quantitative data obtained from the local people responses were analyzed using chi-square test. Qualitative data obtained through focus group discussion and interviews were analyzed by content analysis method.

## Results

### Respondent demographics

The data were collected from September to October, 2022 when the farmers were active in the fields and most of the crop types grown in the area are available on field. Out of the 800 respondents, 464 (58%) were males and 336 (42%) females. The respondent's age ranged from 18 to 70 years of these, the majority were between ages 20 to 59. Further analysis indicated that older farmers (30–70 years old) were most closely associated with reporting the most severe

**Table 1. People's attitude among different age groups towards the existence and conservation of African elephant and the park in Chebera Chrchura National Park, Ethiopia, during year 2022.**

| Age category | Number | Positive | Negative | Neutral |
|---|---|---|---|---|
| 18–20 | 88 | 57.9 | 38.0 | 4.1 |
| 20–29 | 112 | 55.3 | 42.5 | 2.2 |
| 30–39 | 184 | 51.0 | 42.6 | 6.4 |
| 40–49 | 200 | 52.4 | 46.3 | 1.3 |
| 50–59 | 120 | 52.2 | 43.6 | 4.2 |
| >59 | 96 | 53.9 | 41.1 | 5.0 |
| Total | 800 | 53.8 | 42.4 | 3.8 |

level of HEC and had negative attitude towards elephants and their conservation, while younger participants (18–29) showed more positive attitude than other age groups and associated with reporting only minor levels of HEC. In general the majority of respondents, 53.8% showed positive attitude followed by 42.4% who showed negative attitude while, 3.8% were neutral towards the Park and elephant conservation (Table 1). However, this was not significant ($\chi2 = 8.136$, df = 3, P > 0.05).

Of the total 800 respondents, 667 (80.35%) stated their primary occupation was mixed farming (crop cultivation and livestock rearing), few 116 (14.53%) depends on livestock rearing while the remaining 41 (5.11%) non-farmers indicated that their primary occupation was working as daily labors, students or jobless (Table 2).

## Human-elephant conflict

To identify the intensity and types of HEC for farmers living in the elephant conflict area 378 people from the 3 Villages that confirmed the presence of sever HEC in their area were asked

**Table 2. Source of incomes of local communities settled in 20 adjacent villages around Chebera Chrchura National Park, Ethiopia, during year 2022.**

| Village/Kebele | Number | Mixed farming | Livestock | Other |
|---|---|---|---|---|
| Chebra | 43 | 75.1 | 13.2 | 12.7 |
| Seri | 45 | 75.4 | 14.3 | 10.3 |
| Delba | 38 | 79.5 | 15.7 | 4.8 |
| Koyesha | 36 | 60.1 | 12.9 | 27 |
| Oshka | 34 | 81.0 | 17.2 | 1.8 |
| Agare | 38 | 80.2 | 18.3 | 1.5 |
| Kuta | 44 | 82.1 | 14.3 | 3.6 |
| Yora | 46 | 83.0 | 15.1 | 1.9 |
| Shita | 41 | 78.5 | 14.9 | 6.6 |
| Keribella | 35 | 69.7 | 17.8 | 12.5 |
| Menta | 39 | 83.6 | 13.7 | 2.7 |
| Maliga | 38 | 87.0 | 11.1 | 1.9 |
| Dameno | 39 | 86.4 | 12.6 | 1.0 |
| Neda | 44 | 84.3 | 14.5 | 1.2 |
| Churchura | 43 | 83.5 | 14.7 | 1.8 |
| Gudumu | 42 | 79.7 | 16.1 | 4.2 |
| Chawda | 41 | 84.4 | 14.4 | 1.2 |
| Adabacho | 34 | 88.0 | 11.0 | 1 |
| Boka | 41 | 81.8 | 14.7 | 3.5 |
| Gimba | 39 | 84.7 | 14.2 | 1.5 |

**Table 3. The main causes of HEC in the villages where the existence of conflict was confirmed, around CCNP, Ethiopia, during year 2021.**

| Kebles | N of Respondents | Crop raiding | Loss of life stock | Human injury | Loss of human |
|---|---|---|---|---|---|
| Chebra | 126 | 85.8 | 4.7 | 5.5 | 4.3 |
| Seri | 126 | 85.9 | 4.8 | 4.8 | 4.5 |
| Yora | 126 | 86.0 | 5.0 | 4.0 | 5.0 |
| Average | 126 | 85.9 | 4.8 | 4.7 | 4.6 |

Commonly cultivated crops around the villages and palatability to the elephants

about the major cause of HEC, (85.9%) reported crop damage, (4.8%) reported loss of livestock and (4.6%) reported injury and death of human life (Table 3). The difference was statistically significant ($\chi2 = 148.38$, df = 3, P<0.05).

Assessing how local people perceive the risks from HEC related with crop type grown and the severity of crop damage indicated that most of the participants (96%) confirmed that elephants prefer certain crop types over another among the widely cultivated crops Teff, Maize, Banana, Yam and Sorghum were reported being palatable and preferred by the elephants. While spices such as Ginger, Cardamon, and fruits such as Papaya, Mango, Avocado, were confirmed to be unpalatable for the elephants (Table 4).

A total of 61 crop-raiding incidents were recorded during the present study the group structure of crop raiding elephants from the three villages were also recorded. All incidents of crop raid except five occurred during the night. The raids were most frequent (91.9%) during the night (4.9%) of the raid occurred in the evening, while the least frequent the same (1.6%) was occurred during the morning and in afternoon respectively. There were two daytime observations of a mixed group of elephants entering banana farm lands during January 2020 and three bull groups in the evening (Table 5). There was a strong relationship between crop raiding and the time of day.

Local communities used different techniques to control (minimize) the problems caused by elephants on cropland at night. In general most respondents (38%) reported that elephants responded faster for warning gun fired by the Park scouts. They mentioned it as an effective method to prevent crop raid in all villages, several others said the scouts' attempts to ward off

**Table 4. Response of local communities about commonly cultivated crops and palatability to the elephant in the villages, around CCNP, Ethiopia, during year 2021.**

| Village | Chebra | | Seri | | Yora | | Average | |
|---|---|---|---|---|---|---|---|---|
| Total Number | 126 | | 126 | | 126 | | 126 | |
| Response in % | Yes | No | Yes | No | Yes | No | Yes | No |
| Teff | 92 | 8 | 90 | 10 | 92 | 8 | 91.3 | 8.7 |
| Banana | 98 | 2 | 97 | 3 | 98 | 2 | 97.6 | 2. 3 |
| Yam | 97 | 3 | 98 | 2 | 98 | 2 | 97.6 | 2.3 |
| Sorghum | 96 | 4 | 97 | 3 | 94 | 6 | 95.6 | 4.3 |
| Ginger | 4 | 96 | 5 | 95 | 3 | 97 | 4 | 96 |
| Cardamon | 3 | 97 | 2 | 98 | 1 | 99 | 2 | 98 |
| Papaya | 4 | 96 | 4 | 96 | 5 | 95 | 4.3 | 95.7 |
| Mango | 3 | 97 | 2 | 98 | 3 | 97 | 2.7 | 97.3 |
| Avocado | 3 | 97 | 2 | 98 | 2 | 98 | 2.3 | 97.6 |
| Maize | 97 | 3 | 96 | 4 | 99 | 1 | 98 | 2 |

Hour of the day and group structures of crop raiding elephants

**Table 5. Group compositions of elephant herd involved in crop raiding.**

| Hours of the Day | Total Observation | Group structure | Average Number | Percent |
|---|---|---|---|---|
| Morning | 1 | Mixed heard | 39 | 1.6 |
| Afternoon | 1 | Mixed heard | 18 | 1.6 |
| Evening | 3 | Bull Group | 4.5 | 4.9 |
| Night | 56 | Bull Group | 5 | 91.9 |
| Total | 61 | | 5.7 | 100 |

Different HEC prevention/mitigation techniques implemented by the local communities.

the elephants by shooting in the air is effective, but they usually arrive too late. Chilly (19.7%), was mentioned as the second effective method, followed by beehive fence sound noise including the sound of barking dog and hammering materials made of metal (13.3%), guarding (11.3%), fire smoking (9%) and smoking chilly and elephant dung were mentioned as techniques used to chase elephants (Table 5). Views of respondents among villages did not significantly differ ($\chi 2 = 48.82$, df = 6, P>0.05) in using different techniques for protection of crop and livestock. No one used only one method alone but combined and integrated all the local methods to prevent crop raiding by elephants (Table 6).

Overall, traditional methods were not perceived as sufficient to ward off elephants. The house holds interviewed were less positive about the effectiveness of most of the traditional techniques in chasing elephants away from their farm land and believed that elephants has already adapted to most of the traditional techniques. Thus the majority of respondents considered separation of elephants and humans to be the best solution to the problem. Among the respondents 41.8% suggested that barriers such as electric fence; ditch or concrete wall should be constructed in the areas that serve as corridors between the Park boundary and the villages. 25.6% of the respondents wanted to be relocated to other areas that are far enough from the elephant habitats, (20.9%) of the respondents suggested compensation from the government for the crops damaged, (8.5%) suggested killing problem animals that are responsible for the conflict while few of them (3.2%) suggested use of traditional methods of prevention (Table 7). Respondents differed significantly ($\chi 2 = 74.29$, df = 4, P<0.05).

## Focus group discussion with the local community

Focus group discussion showed that in the study area potentially caused economic loss due to livestock depredation and crop damage. In addition to consumption trampling and footing were also observed as a means of crop damage. The discussions held with communities showed that they had negative attitude towards the existence of elephants and the Park. The

**Table 6. Different mitigation techniques implemented by local communities to minimize the crop raiding by elephants around Chebera Chrchura National Park, Ethiopia, during year 2021.**

| Deterrent Techniques | Responses in % | Total Number of respondents |
|---|---|---|
| Guarding | 11.3 | 42 |
| Smoking | 9 | 34 |
| Smoking chilly & elephant dung | 9.7 | 36 |
| Sound Noise | 13.3 | 50 |
| Chilly/beehive fencing | 19.7 | 74 |
| Warning Gun fire | 38 | 142 |
| Total | 100 | 378 |

**Table 7. Recommended HEC measures perceived to be the most effective by local communities around CCNP, Ethiopia, during year 2021.**

| Activity | Frequency | Percentage |
|---|---|---|
| Using traditional method | 12 | 3.2 |
| Shoot them | 32 | 8.5 |
| Compensation | 79 | 20.9 |
| Barrier | 158 | 41.8 |
| Resettlement | 97 | 25.6 |
| Total | 378 | 100 |

discussants stated that the continued existence of elephants had a negative impact on their livelihoods. Few discussants recognized the value of the Park and wildlife for the contribution to the regional economy through tourism and climate stability in the future. Some of the respondents noted that previously they used to hunt and kill elephants and minimize their threat. However, after the establishment of the National Park the area was protected and the population of elephants and their negative effects were increasing from time to time. As a result, some discussants were dissatisfied with the existence of the National Park. They considered the Park as a limiting factor in improving their livelihood. Discussants also stated that the Park has restricted access to resources and caused them forced relocations. Few discussants considered the Park as useless.

They also felt that Park staff members do not like the communities around the Park boundaries and never followed win-win approach. Their main focus was only conserving the wild animals using armed scouts and strong law enforcements. They never considered compensation or any support for families who had lost their family leaders due to HEC however; on the contrary they arrest and bring to courts for disrespecting any rules and regulations of the Park such as killing any wild animal or extraction of resources from the Park. They also believed the situation is beyond their control. Support and government interventions are needed as a way to find a solution for the co-existence of wildlife and communities.

## Discussion

Studies confirmed that benefiting local communities from income generated from protected areas, ecological services provided by wildlife and the recreational values of wildlife protected areas are the main reasons for long-term co-existence between wild animal and local communities living in close proximity with wildlife protected areas [34]. Against this fact frequent conflict with particular animal species may aggravate negative perception about the animal among local communities making the long term co-existence between local communities and the species difficult. In addition to this important factor socio economic variables such as income level, gender, and Age and occupation type highly affect people perception towards wildlife and the species concerned [34].

A good knowledge and understanding of community perceptions of HEC and their attitude towards a particular species and its conservation is vital to ensure long-term co- existence between the species and local communities living in close proximity with wildlife habitats. Managing conflicts is especially challenging for large, potentially dangerous and damaging species such as elephants [28].

In the study area, human-elephant conflict was identified as the top conservation challenge both for the elephant population and local communities living around the Park posing huge problem on the long term co-existence of elephants with the local community. HEC considered as one of the five big issues with equal priority that needs attention regarding the African

elephant conservation [43]. In spite of sever HEC the majority of local communities living adjacent to CCNP had positive attitude for the elephants and supported the conservation of the park and the elephants. The reason for their positive attitude towards both the protected area and the elephants was due to their view about the presence of the National Park and the elephants in their locality as a symbol and unique emblem that attracted both national and international tourists, scientists and higher officials to their area. Similar finding was reported that the residents living around Xishuangbanna Nature Reserve, China have developed positive attitude towards the elephants despite huge damage to their crops due to religious belief [15]. Local communities sharing landscapes with elephants incur a huge cost due to human-elephant conflict that affects the lives and lively hoods of people, imposing negative implications to the households [31]. Elephants were perceived as the most dangerous animals by local communities around CCNP. The level of destruction that the elephant inflict was also claimed to be so much huge. Growing crops free of destruction of elephants is the main concern for most of local communities around the village. Some of the respondents reported that they have forced to abandoning their crops entirely and recently gave up growing crops, in fear of the elephant attack. They mentioned that elephants can destroy the entire field in a single night. These findings are seen in the villages where severe HEC reported from, which are very close to the elephant habitats, coinciding with the previous elephant home range [43]. While, respondents from some other villages confirmed that HEC has never existed in their area. This difference could be due to the villages the location of the villages. Villages located far from the habitats, corridors and home ranges of elephants will experience less conflict than villages that are located near to the elephant habitats. Reports of sever conflict in these areas was also associated with growing crops that were reported to be palatable and attract the elephants and the growing season of these crop types. Some food items/crops were particularly palatable and attract wildlife, Maize and Yam attracted particularly elephants among the crops planted outside the CCNP, Ethiopia [43]. Perceived risk and level of conflict can be reduced by building people's perception towards their ability to avoid negative outcomes through their own actions.

Considering people's perception about human-elephant conflicts, their expectations from mitigation activities and the responsible body to implement activities are vital for effective implementation of the mitigation activities, in addition to the direct physical impacts of the mitigation techniques [45]. Social trust on the capacity of management authorities to implement effective conflict mitigation measures against the elephant attack is therefore among the critical factors that have shaped the perceptions of local communities towards risks associated with the conflicts [45].

As a responsible body for the conservation of elephants and protected areas, which is in charge of controlling any illegal activities committed against the elephant population and the national park, the local communities expect government agencies to protect them from the elephant attack through proper implementation of effective mitigation techniques. The local communities trust in the implementation of different traditional mitigation techniques is generally weak. The local communities were not perceived those methods as sufficient to ward off elephants and they believed that the methods implemented so far were not sufficiently effective. The hose holds interviewed were less positive towards the effectiveness of most of the traditional techniques in chasing elephants away from their farm lands. They believe that elephants has already adapted and do not respond to most of these techniques. Against the above perception in spite of complaints on their usual absence and late arrival, perception of local communities about shooting warning gun by park scouts is among the most accepted effective methods in chasing elephants from their farm lands. Similar finding was reported from other studies that Perception about the capacity of ChNR scouts was more positive and

most respondent's attempts to ward off the elephants by shooting in the air effectively chase the elephants [45].

The majority of respondents believe that separation of elephants and humans by constricting barriers is the best solution to the problem. The idea of constructing barriers such as electric fence; ditch or concrete wall and blocking corridors between the Park boundary and the villages have become the most popular idea of local communities followed by relocating people to other safer places, perceived as the best protection method against the elephant attack irrespective of the associated initial and maintenance costs. The solution favored by most interviewees should be implemented at lease in combination with other mitigation techniques. Failure to implement the solution favored by most interviewees at least in combination of others for different reasons will have negative impacts in the implementation of alternative mitigation methods irrespective of their merits and contributions in minimizing the conflicts [45]. Thus, imposing other alternative techniques without the agreement and active participation and consent, of local communities the communities fail to implement it as a short term or permanent solution. Some individuals may also destruct or steal parts to use for other purposes [28]. The perception of local communities on building barriers as the only reliable conflict mitigation measure might not be realistic to implement in all conflict areas due to the associated high initial and maintenance costs for which the protected area management and the government agencies of Ethiopia have no developed strategic plan and secured budgets so far [43]. In this regard it is recommended to implement a combination of different techniques in different village based on severity of the conflict and suitability of the localities to implement the preferred method [45].

Concerning alternative mitigation measures other than constructing barriers the participants from local communities reported that in spite of their being labor and cost intensive, Chilli and beehive fences are likely to provide better protection. However, these methods require more resource and strong monitoring effort and proper maintenance [45]. Relocating people to other places was another alternative idea forwarded by most interviewees. However it is unrealistic to resettle the entire villages each are with more than 800 households, both cost wise and effectiveness. Moreover at present the original virgin forest in these villages has been completely and permanently removed, and it will no longer serve as food source and shelter for elephants [43].

Understanding the reasoning up on which perceived risks and level of human-elephant conflict laid is critical to address societal or cultural beliefs in order to develop effective mitigation. During focus group discussion Participants indicated that the reason why HEC has become huge problem around their village was done accidental when the elephants were trying to get to food resources. The areas currently facing sever HEC were also confirmed as being the previous common habitats and corridors of elephants before the human settlements and are currently blocked due to the mistakes made in excluding the elephants home range and preferred habitats from the park boundary during demarcation process of the Park in 2005 [43]. However, community members facing HWC in other studies reported they believed that wildlife attacks were motivated primarily by reasons other than ecological drives such as; cultural and superstitions [45].

## Implications for conservation

The CCNP is a very well protected area and known for harboring diversified fauna and flora species, HEC is an increasing concern that needs urgent solution in the area. This study has answered critical issue including the main challenges that the local communities face in the study area and community perceptions and the reasoning up on which perceived HEC and

their attitude towards elephant and its conservation lay and vital to address societal or cultural beliefs in order to develop effective mitigation. Planning and implementation of conservation actions with full consideration of people's perception, their expectations from mitigation activities and the responsible body to implement activities explained by our sample may help effective implementation through strong connection between conservation and human welfare. Future studies that assess the most effective means of including socioeconomic considerations into conservation planning are recommended.

## Supporting information

**S1 File.**
(XLSX)

## Acknowledgments

The authors wish to thank staff members of CCNP, to the local people of Dawuro and Konta Zones and their local administrators for their kind cooperation, patience, hospitality, and willingness to share their knowledge during the interview and discussion about the conservation challenges of the study area.

## Author Contributions

**Conceptualization:** Adane Tsegaye, Afework Bekele, Anagaw Atikem.

**Data curation:** Adane Tsegaye.

**Formal analysis:** Adane Tsegaye.

**Funding acquisition:** Adane Tsegaye.

**Investigation:** Adane Tsegaye.

**Methodology:** Adane Tsegaye.

**Project administration:** Adane Tsegaye.

**Resources:** Adane Tsegaye.

**Software:** Adane Tsegaye.

**Supervision:** Afework Bekele, Anagaw Atikem.

**Validation:** Afework Bekele, Anagaw Atikem.

**Visualization:** Afework Bekele, Anagaw Atikem.

**Writing – original draft:** Adane Tsegaye.

**Writing – review & editing:** Adane Tsegaye, Afework Bekele, Anagaw Atikem.

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
