## [Decision Letter · Decision Letter 0]

8 Aug 2023

PONE-D-23-04990Local Community’s Attitude towards African elephant Conservation in and around

Chebra Churchura National Park, Ethiopia.PLOS ONE

Dear Dr. Tegegn,

Thank you for submitting your manuscript to PLOS ONE. After careful consideration, we feel that it has merit but does not fully meet PLOS ONE’s publication criteria as it currently stands. Therefore, we invite you to submit a revised version of the manuscript that addresses the points raised during the review process.

Authors to please take care the comments / suggestions made by reviewer 1

We look forward to receiving your revised manuscript.

Kind regards,

Muhammad Khalid Bashir, PhD

Academic Editor

PLOS ONE

Journal Requirements:

Funded studies

Funded by Rufford Small Grant

Grant Number: ID:39147-2

Grant Awarded to the coresponding Author ADANE TSEGAYE TEGEGNE

Please state what role the funders took in the study.  If the funders had no role, please state: "The funders had no role in study design, data collection and analysis, decision to publish, or preparation of the manuscript.

No authors have competing interests

5. We note that you have referenced (ie. Bewick et al. [5]) which has currently not yet been accepted for publication. Please remove this from your References and amend this to state in the body of your manuscript: (ie “Bewick et al. [Unpublished]”) as detailed online in our guide for authors

6. We note that Figure 1 in your submission contain map images which may be copyrighted. All PLOS content is published under the Creative Commons Attribution License (CC BY 4.0), which means that the manuscript, images, and Supporting Information files will be freely available online, and any third party is permitted to access, download, copy, distribute, and use these materials in any way, even commercially, with proper attribution. For these reasons, we cannot publish previously copyrighted maps or satellite images created using proprietary data, such as Google software (Google Maps, Street View, and Earth). For more information, see our copyright guidelines: http://journals.plos.org/plosone/s/licenses-and-copyright.

Reviewers' comments:

Reviewer's Responses to Questions

**Comments to the Author**

1. Is the manuscript technically sound, and do the data support the conclusions?

Reviewer #1: Yes

Reviewer #2: Yes

2. Has the statistical analysis been performed appropriately and rigorously? 

Reviewer #1: Yes

Reviewer #2: Yes

3. Have the authors made all data underlying the findings in their manuscript fully available?

Reviewer #1: Yes

Reviewer #2: Yes

4. Is the manuscript presented in an intelligible fashion and written in standard English?

Reviewer #1: Yes

Reviewer #2: Yes

5. Review Comments to the Author

Reviewer #1: The main purpose of the study is to improve understanding of the variables that influence attitudes toward African elephant conservation, with a goal of mitigating conflicts and promoting human-elephant coexistence. The study area is Chebera Churchura National Park Ethiopia (CCNP); located in southwestern part of Ethiopia, in the newly established Southwest Ethiopia Administrative Region (SWER).

The study has following research objectives:

a) to investigate if human-elephant conflicts exist, their main causes and types in different villages.

b) to assess general attitudes of local communities towards elephant conservation and the park management

c) to examine the attitudinal difference towards elephant conservation and the park management among age and sex groups, occupations and educational levels

d) to identify effective HEC mitigation techniques those are trusted by local communities and can be implemented locally

Moreover, the salient features of the current study are:

1) The significance of the study is highlighted in an intelligible way

2) The research problem is genuine as well as has global, regional, and local appeal for wildlife conservation and parks management in the context of global post-2015 development agenda (i.e., SDGs)

3) Estimation od analytical techniques used are clear and appropriate

4) The findings are logically interpreted and discussed in comparison with the previously conducted relevant studies

Some suggestions are here to improve the quality of the manuscript.

1) In the title: the use of the prepositions ‘in’ and ‘around’ requires clarification. Whether the prepositions, respectively imply toward inside the CCNP and outside the CCNP? While in the manuscript, it is explicitly written that for data collection, interviews were conducted from 800 households distributed in 20 park adjacent villages.

2) In the Abstract: Correct spelling of the households (written as hose holds)

3) In the Introduction section, the importance and significance of the research topic (HEC or HWC) could be more highlighted by relating the topic with the relevant sustainable development goals.

4) Given before the section Study Area: correct the spelling of livelihoods (mis-spelled as lively hoods)

5) In the section Questionnaire Design: correct the spelling of chi-square test (written as chi-square taste)

6) In Table 4, correct the spelling of crop teff, (mis-spelled as tefee)

7) Give complete form of the acronyms HEC and HWC at the very first places where the acronyms are used. Subsequently, could be used as abbreviated.

8) In the section Implications for Conservation: use correct acronym CCNP in place of CCNNP

9) Add the originality or novelty of the study

10) Add some details on sampling technique

Reviewer #2: the research is well conducted in a appropriate manner. the objectives and aims of research are clearly exhibited in introduction section. methodologies are clearly explained with all necessary details. Results and discussion section elaborate findings comprehensively.

however few observations that are needed to be revised are as follows:

abstract must be short and crisp. unnecessary details of results in abstract, seems a repetition when is read in result section.

Focus Group Discussions are analysed via content analysis. details of content analysis along with results are missing. you can add the results along side all the results tables or explain them in a separate section.

6. PLOS authors have the option to publish the peer review history of their article (what does this mean?). If published, this will include your full peer review and any attached files.

Reviewer #1: No

Reviewer #2: **Yes: **Saima Afzal Ph. D.

---

## [Author Response · Author response to Decision Letter 0]

1 Sep 2023

Dear Editor,

We have corrected all editorial comments and suggestions and the questions raised by both reviewers are clarified. Specific comments for each reviewer are clarified below:

1. We have cheeked and confirmed that our manuscript meets PLOS ONE's style requirements, including those for file naming. 

2. Financial disclosure: 

Funded studies

Funded by Rufford Small Grant

Grant Number: ID:39147-2

Grant Awarded to the corresponding Author ADANE TSEGAYE TEGEGNE

"The funders had no role in study design, data collection and analysis, decision to publish, or preparation of the manuscript.

3. We declared that no authors have competing interests 

Bewick et al. [5]) which has currently not yet been accepted for publication. Please remove this from your References and amend this to state in the body of your manuscript: (ie “Bewick et al. [Unpublished]”) as detailed online in our guide for authors

4. We want to make a change to our Data Availability statement, all data are included in the manuscript.

5. You said that we have referenced (ie. Bewick et al. [5]) which has currently not yet been accepted for publication, but I did not use this reference in the text or in the reference list

6. we have removed figure 1.

In the figure caption of the copyrighted figure, we have include the following text: “

Reprinted from [AGBIR-22-73114] under a CC BY license, with permission from [Adane Tsegaye], original copyright [2022].”

7. We have reviewed the reference list to ensure that it is complete and correct and confirmed it is complete and correct. 

Response to comments from reviewer 1

1. Concerning the prepositions ‘in’ and ‘around’ in the title Local People’s Attitude towards African elephant Conservation in and around Chebra Churchura National Park, Ethiopia. Even though the data collection, interviews were conducted from 800 households distributed in 20 park adjacent villages totally out of the park boundary our intension includes investigating the local communities attitude if it is totally against elephant conservation whether it is inside the park or outside. We want to answer the question if it is it the elephants attack in the surrounding communities that shaped their attitude towards the elephant conservation. If that is the case they will have positive attitude towards the elephant conservation as long as the elephants do not come out of the park and attack them. On the other hand if their attitude is not attached to the elephants attack they will develop the same positive or negative attitudes irrespective of the elephants attack so that they will not be interested in implementing HEC mitigation measure. However, if using the preposition in do not sounds good and make the title explanatory we are willing to remove the preposition “in” from it.

2. We have cheeked and corrected the spelling of the households (written as hose holds) in the Abstract.

3. In the Introduction section, we added one additional paragraph that explain about the importance and significance of the research topic (HEC or HWC) by relating the topic with the relevant sustainable development goals.

4. We have corrected the spelling of livelihoods (mis-spelled as lively hoods) in the section given before the section Study Area:

5. In the section Questionnaire Design we have corrected the spelling of chi-square test (written as chi-square taste).

6. In Table 4, we have correct the spelling of crop teff, (mis-spelled as tefee),

7. We have given complete form of the acronyms HEC, CCNP and HWC at the very first places where the acronyms are used. Subsequently, we used as abbreviated throughout the manuscript.

8. In the section Implications for Conservation: we have corrected acronym CCNP in place of CCNNP.

9. We added a text that explains about the originality or novelty of the study just before the acknowledgment section.

10. Sampling technique was clarified

Response to Reviewer 2.

1. Abstract must be short and crisp. We have removed a paragraph which was unnecessary details of results in abstract which seems a repetition when is read in result section. We removed 4 lines between line 14 and 19 which was described as “Most respondents (38%) reported firing warning gun by park scouts as an effective method of crop prevention followed by Chilly and bee hive fencing (19.7%) sound noise including the sound of barking dog and hammering materials made of metal (13.3%) guarding (11.3%) fire smoking (9%) and smoking chilly and elephant dung (9%)”

2. Based on the reviewers comment we explained the result in the separate section “entitled” focus group discussion.

---

## [Decision Letter · Decision Letter 1]

26 Sep 2023

Local Community’s Attitude towards African elephant Conservation in and around

Chebra Churchura National Park, Ethiopia.

PONE-D-23-04990R1

Dear Dr. Tegegn,

We’re pleased to inform you that your manuscript has been judged scientifically suitable for publication and will be formally accepted for publication once it meets all outstanding technical requirements.

Kind regards,

Muhammad Khalid Bashir, PhD

Academic Editor

PLOS ONE

Additional Editor Comments (optional):

Reviewers' comments:

Reviewer's Responses to Questions

**Comments to the Author**

1. If the authors have adequately addressed your comments raised in a previous round of review and you feel that this manuscript is now acceptable for publication, you may indicate that here to bypass the “Comments to the Author” section, enter your conflict of interest statement in the “Confidential to Editor” section, and submit your "Accept" recommendation.

Reviewer #1: All comments have been addressed

Reviewer #2: All comments have been addressed

2. Is the manuscript technically sound, and do the data support the conclusions?

Reviewer #1: Yes

Reviewer #2: Yes

3. Has the statistical analysis been performed appropriately and rigorously? 

Reviewer #1: Yes

Reviewer #2: Yes

4. Have the authors made all data underlying the findings in their manuscript fully available?

Reviewer #1: Yes

Reviewer #2: Yes

5. Is the manuscript presented in an intelligible fashion and written in standard English?

Reviewer #1: Yes

Reviewer #2: Yes

6. Review Comments to the Author

Reviewer #1: In response to the previously proposed suggestions, the authors have incorporated all the suggestions. Undoubtedly, the article is suitable for publications in this current form.

1. Authors have now better justified the use of the preposition in the title of the paper

2. In the Abstract, the spelling of the ‘household’ are corrected.

3. By adding an extra paragraph in the Introduction section, the importance and significance of the research topic (HEC or HWC) has been linked to the relevant sustainable development goals.

4. Authors have corrected the spelling of livelihoods (mis-spelled as lively hoods) in the section given before the section Study Area

5. In the section ‘Questionnaire Design’, authors have corrected the spelling of chi-square test (written as chi-square taste).

6. In Table 4, the spelling of crop ‘teff’, (mis-spelled as tefee) are corrected

7. Complete form of the acronyms HEC, CCNP and HWC are added at the very first places where the acronyms are used.

8. In the section ‘Implications for Conservation’ corrected acronym CCNP is added

9. The originality or novelty of the study is also incorporated.

10. Sampling technique has been clarified

Reviewer #2: (No Response)

7. PLOS authors have the option to publish the peer review history of their article (what does this mean?). If published, this will include your full peer review and any attached files.

Reviewer #1: No

Reviewer #2: **Yes: **Dr. Saima Afzal

---

## [Editor Report · Acceptance letter]

18 Oct 2023

PONE-D-23-04990R1 

Local Community’s Attitude towards African elephant Conservation in and around Chebra Churchura National Park, Ethiopia. 

Dear Dr. Tsegaye:

I'm pleased to inform you that your manuscript has been deemed suitable for publication in PLOS ONE. Congratulations! Your manuscript is now with our production department. 

Kind regards, 

on behalf of

Dr. Muhammad Khalid Bashir 

Academic Editor

PLOS ONE